# The Effects of Increasing Fruit and Vegetable Intake in Children with Asthma on the Modulation of Innate Immune Responses

**DOI:** 10.3390/nu14153087

**Published:** 2022-07-27

**Authors:** Banafsheh Hosseini, Bronwyn S. Berthon, Megan E. Jensen, Rebecca F. McLoughlin, Peter A. B. Wark, Kristy Nichol, Evan J. Williams, Katherine J. Baines, Adam Collison, Malcolm R. Starkey, Joerg Mattes, Lisa G. Wood

**Affiliations:** 1Priority Research Centre for Healthy Lungs, Hunter Medical Research Institute, University of Newcastle, Newcastle, NSW 2305, Australia; b.hosseini.bh@gmail.com (B.H.); bronwyn.berthon@newcastle.edu.au (B.S.B.); bec.mcloughlin@newcastle.edu.au (R.F.M.); peter.wark@health.nsw.gov.au (P.A.B.W.); kristy.nichol@newcastle.edu.au (K.N.); evan.j.williams@newcastle.edu.au (E.J.W.); katherine.baines@newcastle.edu.au (K.J.B.); adam.collison@newcastle.edu.au (A.C.); malcolm.starkey@newcastle.edu.au (M.R.S.); 2Priority Research Centre Grow Up Well, Hunter Medical Research Institute, University of Newcastle, Newcastle, NSW 2305, Australia; megan.jensen@newcastle.edu.au (M.E.J.); joerg.mattes@newcastle.edu.au (J.M.); 3Department of Respiratory and Sleep Medicine, John Hunter Hospital, Newcastle, NSW 2305, Australia; 4Department of Immunology and Pathology, Central Clinical School, Sub-Faculty of Translational Medicine and Public Health, Monash University, Melbourne, VIC 3004, Australia

**Keywords:** childhood asthma, innate immunity, fruit and vegetables, carotenoids

## Abstract

Children with asthma are at risk of acute exacerbations triggered mainly by viral infections. A diet high in fruit and vegetables (F&V), a rich source of carotenoids, may improve innate immune responses in children with asthma. Children with asthma (3–11 years) with a history of exacerbations and low F&V intake (≤3 serves/d) were randomly assigned to a high F&V diet or control (usual diet) for 6 months. Outcomes included respiratory-related adverse events and in-vitro cytokine production in peripheral blood mononuclear cells (PBMCs), treated with rhinovirus-1B (RV1B), house dust mite (HDM) and lipopolysaccharide (LPS). During the trial, there were fewer subjects with ≥2 asthma exacerbations in the high F&V diet group (n = 22) compared to the control group (n = 25) (63.6% vs. 88.0%, *p* = 0.049). Duration and severity of exacerbations were similar between groups. LPS-induced interferon (IFN)-γ and IFN-λ production showed a small but significant increase in the high F&V group after 3 months compared to baseline (*p* < 0.05). Additionally, RV1B-induced IFN-λ production in PBMCs was positively associated with the change in plasma lycopene at 6 months (r_s_ = 0.35, *p* = 0.015). A high F&V diet reduced asthma-related illness and modulated in vitro PBMC cytokine production in young children with asthma. Improving diet quality by increasing F&V intake could be an effective non-pharmacological strategy for preventing asthma-related illness by enhancing children’s innate immune responses.

## 1. Introduction

Asthma is the most prevalent chronic childhood disease, ranking among the top 20 conditions worldwide for disability-adjusted life years in children [1]. Children with asthma frequently experience exacerbations [2]. Acute asthma exacerbations are the primary cause of urgent healthcare visits, hospitalisations, mortality, and incur significant treatment costs [2]. Respiratory virus infections are the most common cause of asthma exacerbations in children [3], with rhinoviruses (RVs) being the most common cause of virus-associated exacerbations in children over the age of 3 years [4]. RV infection provokes asthma symptoms and can also impair lung function in patients with asthma [5]. Other environmental factors such as bacterial infection and allergen exposure play an essential role in asthma exacerbations. Several studies have shown a consistent relationship between levels of household lipopolysaccharide (LPS) and house dust mite (HDM) exposure [6] with asthma exacerbations [7,8,9]. 

Asthma is usually managed using inhaled corticosteroids (ICS); however, viral-induced exacerbations can occur despite ICS treatment in children with asthma [10,11]. One proposed non-pharmacological, adjuvant therapeutic approach to prevent and manage asthma is increasing the dietary intake of fruit and vegetables, a strategy that has been reported by the European Academy of Allergy and Clinical Immunology (EAACI) to decrease asthma incidence, particularly in children [12]. Fruit and vegetables are high in antioxidants and anti-inflammatory phytochemicals, including carotenoids (e.g., lutein, lycopene, β-cryptoxanthin, α-carotene and β-carotene) and other biologically active substances [13]. Dietary antioxidants can scavenge reactive oxygen species (ROS) [14], which are increased in the airways of patients with asthma [15], and thus, inhibit nuclear factor-kappa B (NFκB)-mediated inflammation [14]. However, this capacity is reduced following a low F&V diet [16].

Many epidemiological studies have shown that total fruit and vegetable (F&V) intake is inversely associated with the risk of asthma [17,18] and wheezing [19,20,21] and is positively associated with lung function [22,23]. Indeed, our previous meta-analyses showed inverse associations between fruit consumption and risk of prevalent wheeze and asthma severity [24]. Similarly, vegetable intake was inversely associated with the risk of asthma [24]. Extending these observations, in our previous randomised control trial (RCT) conducted in 137 adult patients with asthma, we demonstrated that those assigned to a low versus a high F&V diet (<3 vs. ≥7 serves of F&V per day) for 14 weeks had a 2.26-fold increased risk of an asthma exacerbation [25].

This study reports secondary findings from an RCT that delivered a 6-month dietary intervention to increase F&V intake in children with asthma [26]. We have previously demonstrated that high F&V diet for 6 months did not prevent exacerbations, though it was associated with improvements in lung function [26]. In this study, we aimed to extend our previous findings, hypothesising that a high F&V diet would protect children with asthma from exacerbations via enhancing innate immune responses.

## 2. Materials and Methods

### 2.1. Study Design and Participants

The complete study design and methodology have been described previously [26]. Participants were recruited via attendance to the emergency department or admission to the John Hunter Children’s Hospital, Newcastle, Australia, and Maitland Hospital, Newcastle, Australia, following an asthma exacerbation, from September 2015 to July 2018.

Children aged 3–11 years were eligible if they had a physician diagnosis of asthma; recent exacerbation/s (≥1 exacerbation in the preceding 6 months or ≥2 in the past 12 months); stable asthma at the initial clinic visit (defined as no change in asthma medications, unscheduled medical visit for asthma, use of OCS or antibiotics in preceding 4 weeks); consuming ≤3 serves of F&V per day (assessed over the past week); willingness and ability to attend clinic appointments; desire to comply with proposed dietary changes; and agreement to collect blood samples for research purposes at clinic visits. Exclusion criteria included other respiratory conditions, diagnosed intestinal disorders, or consumption of nutritional supplements (in the previous 4 weeks).

Subjects randomly assigned to the high F&V diet were encouraged to meet age-appropriate Australian Dietary Guideline (ADG) recommendations for F&V serves/day [27]. The participants in the control group, blinded to the study hypothesis, continued their usual diet (≤3 serves/d F&V).

Food hampers were provided according to group allocation: the high F&V group received fortnightly food hampers, which included a selection of fresh and frozen fruit and vegetables, while the control group received monthly food baskets, which included a selection of carbohydrate-based foods, low in soluble fibre and antioxidants (bread, rice, pasta, cereal). The hampers were tailored to the subject’s preferences, within the requirements of the dietary intervention.

Adherence was assessed by 24-h food recalls [27] collected at each clinic visit and during fortnightly telephone calls.

Children were screened to ensure they did not have any symptoms of cold/flu at the time of sample collection. The study was approved by the Hunter New England Ethics Committee (15/06/17/4.03) and registered with the University of Newcastle Human Research Ethics Committee. Written informed parental consent, and child assent (where applicable) was obtained before participation in the study. The children and their families were given an information brochure on general healthy eating for their age and a personalised consultation with a dietitian at the end of the trial.

### 2.2. Outcomes

The primary outcomes of this trial have been reported previously [26]. The secondary outcomes of this study included frequency of respiratory-related adverse events, severity and duration of asthma exacerbations, viral detection in nasal swabs collected during asthma-related events, and in-vitro cytokine production in PBMCs treated with different stimuli.

### 2.3. Clinical Assessment

Detailed methods of clinical assessments are reported elsewhere [26]. Briefly, at baseline, 3 months (±2 weeks) and 6 months (±2 weeks), all participants fulfilling the inclusion criteria attended the clinic at the Hunter Medical Research Institute, Newcastle, Australia, for clinical assessment and blood collection following a 12-h overnight fast.

### 2.4. Asthma Exacerbations and Upper Respiratory Tract Infection

Parent-reported asthma-related illness was categorised into three groups, based on the child’s symptoms and signs: (1) exacerbation: a parent-reported asthma exacerbation alone without the presence of upper respiratory tract infection (URTI) symptoms; (2) URTI: a parent-reported URTI alone (runny/congested nose, sore throat, earache, sneezing, with or without fever), with no change in asthma symptoms; and (3) exacerbation with URTI: a parent-reported increase in cough, wheeze or shortness of breath with suspected URTI.

At the time of a suspected URTI or asthma exacerbation, the parent/guardian was asked to complete the validated Asthma flare-up diary for young children (ADYC) questionnaire to document the presence, duration and severity of an asthma exacerbation [28]. Parents were also instructed to collect a nasal swab at the onset of an asthma exacerbation or a suspected URTI (no later than day one or two of symptoms). For additional details, please see this article’s Appendix A.

### 2.5. Laboratory Methods

#### 2.5.1. Carotenoid Analysis

High-Performance Liquid Chromatography (HPLC) was used to measure plasma carotenoid concentrations using Agilent 1200 Series HPLC with Chemstations software (Agilent Corporation, Waldbronn, Germany) as described previously [29,30].

#### 2.5.2. Identification of Respiratory Viruses in Nasal Samples

Viral RNA was extracted from 140 µL of each nasal sample using the QIAamp Viral RNA Mini Kit (Qiagen, Australia), following the manufacturer’s instructions [31]. Samples were then stored at −80 °C until the qPCR assays were performed. For additional details, please see this article’s Appendix A.

#### 2.5.3. PBMC Isolation and Culture

PBMCs were isolated from whole blood by density gradient method [32] using Ficoll-PaqueTMPLUS (GE Healthcare, Sydney, Australia) and cultured with and without RV1B, LPS or HDM for 48 h. For additional details, please see this article’s Appendix A.

#### 2.5.4. Cytokine Assays

Cell culture supernatant concentrations of IFN-γ, IL-1β and IL-6, were analysed using a bead-based multiplex assay (BD Bioscience, Sydney, Australia), and IFN-λ concentrations were measured using high-sensitivity commercial ELISA assays (R&D Systems, Sydney, Australia) as per the manufacturer’s recommendations. For additional details, please see this article’s Appendix A.

### 2.6. Statistical Analysis

To evaluate the true, undiluted, effect of the 6-month F&V intervention, a complete case per-protocol analysis (PPA) was completed, excluding intervention group subjects who consumed below ADG F&V serves in ≥four 24-h food recalls (n = 2). Data were reported as mean ± standard deviation or median [interquartile range]. Significant differences between groups were determined using an independent *t*-test (parametric data), the Mann–Whitney U-test, or the Wilcoxon signed-rank test as appropriate (non-parametric data). Group differences in change from baseline during the intervention were analysed using linear mixed models (LMMs) with group (intervention or control) and time (treated as categorical with levels at baseline (0 months), 3 and 6 months). LMMs were adjusted for the number of hospitalisations in the previous 12 months, and random effects were specified for time. LMMs use all data available at each time point; therefore, missing data imputation was not undertaken. Group differences in ADYC scores from baseline during the intervention were analysed using LMM adjusted for maintenance ICS use, accounting for multiple events per child. Pearson’s chi-squared and Fisher’s exact tests (two-tailed) were used to compare virus detection rates in nasal samples from both groups. Linear correlation between variables was measured using the Pearson product-moment correlation coefficient (r) or Spearman rank correlation (r_s_), as per distribution. Statistical analyses were performed using STATA 15 (StataCorp, College Station, TX, USA). Significance was accepted when *p <* 0.05.

## 3. Results

### 3.1. Subjects’ Characteristics

Sixty-seven children (median age of 5 (3, 7) years) with stable asthma were randomised to the high F&V diet (n = 33) or control group (n = 34). Of these, 16 subjects were either lost to follow-up or withdrew, with two excluded due to non-adherence to the dietary intervention. Forty-seven participants (intervention n = 22, control n = 25) were included in the per-protocol analysis (Appendix A).

Compared to the control group, significantly more individuals in the intervention group were exposed to tobacco smoke in-utero (*p* = 0.015), and they had ≥1 hospital admission for asthma in the previous 12 months (*p* = 0.027); all other baseline characteristics did not differ between groups (Table 1). Medication use in the last 12 months was also similar between groups (Table 2).

### 3.2. Changes in Fruit and Vegetable Intake and Plasma Carotenoids

Efficacy of intervention has been reported previously [26]. F&V consumption over the trial duration was significantly higher in the intervention group than in the control group (*p* < 0.001), with no significant between-group difference detected at baseline. Similarly, after 3 and 6 months, there was a significant change in plasma carotenoid levels (as an objective biomarker of F&V intake) between groups, with no significant between-group difference observed at baseline [26].

### 3.3. Frequency and Severity of Asthma-Related Events

The high F&V diet group had significantly fewer subjects with 2 or more asthma exacerbations and URTI during the 6-month intervention (88.0% versus 63.6%, *p* = 0.049; Table 3). The frequency of reported exacerbations, URTI, and exacerbations with URTI was similar between the two groups (*p* > 0.05).

A total of 135 ADYC diaries were completed and returned (intervention n = 20, 73 events; control n = 22, 62 events) with 71 valid ADYC exacerbation events included in analysis (intervention n = 18, 34 events; control n = 18, 37 events). No significant group difference was observed in the severity, duration, or use of β2-agonists or OCS for exacerbations recorded by parents using the ADYC (Table 4).

### 3.4. Viral Detection in Nasal Swabs Collected during Asthma-Related Events

In total, 45 respiratory-related adverse events in the intervention group and 56 events in the control group were assessed by qPCR for common respiratory viruses. PCR analysis was positive for one or more respiratory viruses in 16 (35.6%) of the total events in the intervention group and 21 (37.5%) of the total events in control subjects. Overall, there was no difference between groups in virus detection rates for asthma exacerbations or URTIs. Rhinovirus was the most common respiratory virus detected during an event in the intervention (22.2%) and control (28.6%) groups. The second most prevalent virus during an event was Influenza B (6.6%) in the intervention group and Influenza A for the control group (9.0%). The results of the qPCR tests for each event are reported in (Table 5).

### 3.5. In Vitro PBMC Cytokine Production

No significant between-group difference was observed after stimulation of PBMCs at baseline (Table 6). However, after 3 months, LPS-induced IFN-γ and IFN-λ production showed a small but significant increase in the intervention group compared with baseline. No significant within or between-group changes were observed in LPS-induced IL-1β and IL-6 production in PBMCs during the 6-month intervention. There were no significant within or between-group changes in RV1B-induced IFN-γ, IFN-λ, IL-1β, and IL-6 production in PBMCs during the trial. Similarly, no significant within or between-group changes were observed in HDM-induced IFN-γ, IFN-λ, IL-1β, and IL-6 production in PBMCs over the duration of the study.

Overall, our model revealed that IL-1β and IL-6 production in PBMCs were more sensitive to LPS stimulation than HDM and RV1-B. In contrast, LPS and HDM stimulations were weak inducers of IFN-γ and IFN-λ compared to RV1B. This trend was similar in both groups at all time points studied.

### 3.6. Associations

The associations between PBMC cytokine responses and plasma carotenoid levels at 6 months showed a positive correlation between RV1B-induced IFN-λ production and change in plasma lycopene (r_s_ = 0.35, *p* = 0.015)**.** RV1B-induced IL-6 production was inversely associated with plasma lutein levels (r_s_ = −0.29, *p* = 0.045). Furthermore, HMD-induced IL-6 production was inversely correlated with change in plasma lutein (r_s_ = −0.28, *p* = 0.045). Similarly, inverse associations were identified between LPS-induced IL-1β production and change in plasma α-carotene (r_s_ = −0.32, *p* = 0.025), and β-carotene (r_s_ = −0.33, *p* = 0.020), as well as total carotenoids (r_s_ = −0.34, *p* = 0.016).

## 4. Discussion

This paper investigates the adverse respiratory events and innate immune responses of children with asthma following a 6-month high F&V dietary intervention. The intervention modified asthma-related illness, as the high F&V diet group had significantly fewer subjects with ≥2 asthma exacerbations/respiratory infections than the control group during the trial. Plasma carotenoids increased in the high F&V group, which correlated with self-reported F&V intake. LPS-induced IFN-γ and IFN-λ production was significantly increased in the intervention group after 3 months, compared to baseline; however, at the end of the 6-month trial, no significant within or between-group changes were observed in cytokine production from PBMCs stimulated with RV1B and HDM.

Previously our research group observed that when adults with asthma were assigned to a high F&V diet (≥7 serves/day) and compared with those assigned to a low F&V diet (<3 serves/day), the low F&V diet group were 2.26 times more likely to exacerbate [25]. In the present study, we have also observed an effect of a high F&V diet on respiratory events, as the number of subjects who experienced at least two episodes of asthma exacerbation and URTI during the trial was significantly lower (63.6%) in the high F&V diet group compared with the control group (88%). Overall, the number of reported asthma-related events was lower in the intervention compared to the control group; however, this trend did not reach statistical significance. Further, there were no significant group differences in the severity and duration of asthma exacerbations recorded by parents using the ADYC. In this study, self-completion ADYC diaries were used to capture information about the onset and severity of exacerbation. Reliance on self-completed journals may not be suitable for evaluating exacerbation severity in clinical trials as they are not always completed correctly or returned. In our study, eleven (23.4%) parents did not return valid diaries because of the absence of episodes and poor compliance. Future RCTs should consider evaluating whether other methods, such as an online version and electronic reminders, would improve compliance. Additionally, parental perception of symptom severity can affect the scores [28]. Evidence shows that parents can have misperceptions about asthma and its management, as they tend to underestimate the severity of their child’s asthma and overestimate asthma control [33,34].

There was no significant difference in virus detection in our study, with a virus detection rate of just 35.6% in the intervention group and 37.5% in control subjects. Previous studies reported higher rates of respiratory virus detection, ranging from 40% to 60% after asthma exacerbation in children [11,35]. In the present study, we focused on testing for a single virus of interest for each assay, whereas the studies with higher detection rates [11,35] used a gene expression panel that tests for multiple respiratory pathogens simultaneously. Moreover, in the study by Khetsuriani et al. [35], viruses were detected in 60% of patients using a combination of throat swabs and nasal samples. Therefore, it can be suggested that the difference in viral detection rates may be due to the sample collection method [35] and the differences in the use of PCR assays. Nonetheless, RV was the most prevalent respiratory virus detected in samples from the intervention (22.2%) and control (28.6%) groups. This is consistent with previous studies that reported that RVs are the most common precipitants of virus-associated exacerbations in children over the age of 3 years [4,11,36].

Cytokine responses in PBMCs stimulated with RV1B, LPS and HDM were also measured, with increased IFN-γ and IFN-λ production in LPS-exposed PBMCs from intervention group subjects at 3 months compared to baseline. IFNs can interact with specific cellular receptors and induce innate and adaptive immune responses, which are crucial for mediating host defences against viral and bacterial infections [37]. IFN-γ is one of the critical mediators of LPS-induced immune responses [38]. IFN-γ intensifies antimicrobial immune responses via inducing macrophage functions such as phagocytosis, respiratory burst activity, antigen presentation, and cytokine secretion. The functional significance of IFN-γ in antimicrobial defence is indicated by the increased susceptibilities of IFN-γ^−/−^ and IFN-γR^−/−^ mice to a wide range of infections [38]. We have previously shown that PBMCs from children with asthma had deficient IFN-γ production in response to both RV1B and LPS infection compared with healthy controls [39]. Other studies have also revealed that children with asthma, compared with age-matched controls, had defective or impaired IFN responses to respiratory viruses [40,41]. Therefore, an increase in IFN production in children with asthma could be expected to protect them from inflammatory injury.

To our knowledge, this is the first study to examine the effects of a high F&V diet on innate immune responses of PBMCs stimulated with RV1B, LPS and HDM in children with asthma. Two intervention studies [42,43] have examined the effect of withdrawal of antioxidant-rich foods (in particular F&V) from the diet in adults with asthma. Antioxidant withdrawal resulted in increased airway neutrophils [43] and upregulation of inflammatory and immune response genes in sputum cells, including the innate immune receptors TLR2, IL1R2, CD93, ANTXR2, the innate immune signaling molecules IRAK2, IRAK3, MAP3K8 and neutrophil proteases MMP25 and CPD [42]. Other research has mostly focused on the effects of isolated nutrients on immune responses in asthma. For example, in a murine model of asthma, administration of lycopene significantly increased IFN-γ expression [44]. Saedisomeolia et al. [45] showed that pre-incubation of airway epithelial cells with lycopene reduced the release of IL-6 following exposure to LPS. Similarly, in a mouse model of allergic airway inflammation [46], lycopene supplementation decreased allergen-induced release of the T helper 2–associated cytokines IL-4 and IL-5. In another study [47], pre-treatment of PBMCs with lutein was shown to suppress LPS- induced mRNA expression of IL-6 and IL-1β. Therefore, combined with the evidence for a whole-food intervention, it can be concluded that dietary antioxidants induce protective innate immune responses in asthma.

Correlation analysis of PBMC cytokine responses to RV1B, HDM, and LPS stimulation and plasma carotenoid levels showed that changes in total and individual carotenoids (such as lycopene, lutein, α- and β-carotene) were associated with higher IFN-λ production, while they were inversely associated with IL-6 and IL-1β production. Previous research has explored the association between dietary carotenoid intake and innate immune mediators. No association was found between intake of β-carotene and in vitro production of LPS-induced cytokines, including IL-1β, IL-6, IL-8, IL-10 and tumour necrosis factor (TNF)-α [48]. One explanation for this discrepancy may relate to the methods used for assessing carotenoid status. In the study by Vivek [48], dietary intake of carotenoids was evaluated using a food frequency questionnaire, whereas, in our study, we used an objective approach of directly measuring plasma carotenoid levels, thus providing a more accurate indication of in vivo carotenoid concentrations, which is a strength of this study. It would be reasonable to suggest that the protective effects of plasma carotenoid levels on innate immune responses observed in our study could be due to the consumption of a high F&V diet. Further research is needed to investigate and confirm this.

Findings from the Australian National Health Survey indicated that only 5.1% of all children met the recommended daily intake for F&V [49]. These trends are of public health concern. While cost may be a significant barrier to fruit and vegetable consumption, other factors such as taste, preference, and culture can also significantly impact F&V purchase and consumption [50]. There is growing evidence to indicate that participation in incentive programs such as community kitchens or school gardens can help promote consumption of F&V [51,52]. In our study, the intervention more than doubled F&V intake in children with asthma. Parental and children’s knowledge of the protective effect of the F&V consumption on asthma is of great importance and needs to be included in the management of childhood asthma.

The current study has some limitations. Blinding of the treatment allocation was not feasible for this study [53]. However, this is typical of many dietary intervention studies. The participants were blinded to the study hypothesis to reduce bias, and a range of objective outcomes was included in the assessments. Furthermore, the attrition rate (26.8%) was high, which may be related to the parent burden, including time and commitment associated with study visit attendance, completing study questionnaires and participation in phone consultations and/or child discomfort during blood collection. However, this was expected and is comparable to other intervention studies in children with asthma [54,55]. Finally, due to insufficient PBMC samples, we could not determine whether cytokines responses would differ at 24 h or 72 h. However, a deficient interferon response to RV in asthmatic children was also reported in a previous study that harvested supernatants at 72 h [56].

## 5. Conclusions

In summary, we showed that increasing F&V intake decreased asthma-related illness. This study establishes the relevance of a dietary intervention strategy—increased F&V intake—for improving innate immune responses and reducing exacerbation rates in children with asthma. This is an appealing strategy with no side effects, which is likely to be widely accepted and adopted by children and their caregivers.

## Figures and Tables

**Table 1 nutrients-14-03087-t001:** Baseline demographics and clinical characteristics.

Baseline Characteristic	Intervention n = 22	Control n = 25	*p*
Age (years), median (range)	5 (3–10)	5 (3–11)	0.827
Age 3–6 years, n (%)	17 (77)	18 (72)	0.747
Age 7–11 years, n (%)	5 (23)	7 (28)	
Sex (Male: Female)	15:7	19:6	0.550
Race: White, n (%)	17 (77.3%)	20 (80.0%)	0.303
Height (cm), mean ± SD	117 ± 14	116 ± 16	0.988
Weight (kg), median (IQR)	21.6 (16.9, 25.1)	21.4 (16.9, 26.6)	0.664
BMI z-score, mean ± SD	0.1 ± 1.3	0.1 ± 1.4	0.922
BMI percentile, mean ± SD	49.4 ± 32.8	54.7 ± 32.4	0.623
**Risk factors**, n (%)			
Current food allergy	6 (27)	5 (20)	0.557
History of Eczema ^#^	16 (73)	12 (48)	0.085
History of Hay fever ^	16 (73)	12 (48)	0.085
Asthma in first degree relative *	15 (68)	16 (64)	0.592
Maternal Asthma	7 (32)	8 (32)	0.923
Paternal Asthma	6 (27)	10 (40)	0.418
Family history of Eczema	11 (52)	17 (68)	0.280
Family history of Hay fever	18 (86)	21 (84)	0.872
In-utero tobacco exposure ^†^	5 (23)	0	**0.015**
Passive smoke exposure at home	3 (14)	1 (4)	0.318
**Morbidity in previous 12 months**			
ED visits for asthma, mean ± SD	1.7 ± 1.0	1.6 ± 0.9	0.672
≥1 hospital admission, n (%)	16 (73)	10 (40)	**0.027**
Hospitalisations, median (IQR)	1 (0,1)	0 (0, 1)	0.142

BMI z-scores and percentiles were calculated with reference to the Centre for Disease Control and Prevention 2000 Growth Charts. ^#^ Based on parental response to “Has your child ever had eczema?” ^ Based on parental response to “Has your child ever had a problem with sneezing, or a runny or blocked nose when he/she DID NOT have a cold or the flu?”. * Based on parental response to “Has anyone in the child’s immediate family ever had asthma? (Including mother, father or direct siblings)”. ^†^ Based on parental response to “Does anyone living in the child’s immediate home (where the child spends more than half of his/her time) smoke, even if he/she smokes outside?” Difference between groups analysed by the Wilcoxon Rank sum test (non-parametric data), two-sample *t*-test (parametric data) or Pearson’s Chi-squared test/Fisher’s exact test (testing equality of proportions) where appropriate. *p* < 0.05 considered statistically significant. Values in bold indicate statistically significant results. Abbreviations: IgE, immunoglobulin E; IQR, interquartile range; BMI, body mass index; ED, emergency department.

**Table 2 nutrients-14-03087-t002:** Medication at baseline and in the previous 12 months.

Type of Medication	Intervention n = 22	Control n = 25	*p*-Value
Short courses of OCS, median (IQR)	3 (1, 4)	2 (1, 4)	0.974
≥2 Short courses of OCS, n (%)	17 (68)	13 (59)	0.526
ICS or ICS/LABA ever, n (%)	16 (73)	14 (56)	0.234
ICS intermittent, n (%) *	1 (5)	5 (20)	0.194
ICS maintenance, n (%) ^	13 (59)	8 (32)	0.062
ICS/LABA maintenance, n (%) ^	2 (9)	1 (4)	0.593
ICS dose, beclomethasone equiv., median (IQR)	400 (200, 400)	400 (200, 500)	0.573
Montelukast, n (%)	4 (18)	5 (20)	1.000
SABA only, n (%)	4 (18)	9 (36)	0.207
Intranasal CS, n (%)	3 (14)	3 (12)	1.000

^ Reported to have been taken for most of the previous 12 months. Difference between groups analysed by the Wilcoxon Rank sum test (non-parametric data) or Pearson’s Chi-squared test/Fisher’s exact test (testing equality of proportions) where appropriate. *p* < 0.05 considered statistically significant. Abbreviations: IQR, interquartile range; OCS, oral corticosteroids; ICS, inhaled corticosteroids; LABA, long-acting β2-agonist; SABA; short-acting β2-agonist; CS, corticosteroid. * Reported to have been taken intermittently or on an as-needed basis.

**Table 3 nutrients-14-03087-t003:** Frequency of asthma exacerbations and upper respiratory tract infections reported by parent during the 6-month intervention study.

Event Type		Intervention (n = 22)		Control (n = 25)		*p*-Value
		n	%	n	%	
**All Events (1+2+3)**	**None**	2	9.0%	0	0%	0.214
	**1 or more**	20	91.0%	25	100%	0.214
	**2 or more**	14	63.6%	22	88%	**0.049**
**Exacerbations (1)**	**None**	12	54.6%	12	48%	0.654
	**1 or more**	10	45.4%	13	52%	0.654
	**2 or more**	5	22.7%	3	12%	0.446
**URTI (2)**	**None**	14	63.6%	13	52%	0.421
	**1 or more**	8	36.4%	12	48%	0.421
	**2 or more**	3	13.6%	6	24%	0.470
**Exacerbations with URTI (3)**	**None**	5	22.7%	4	16%	0.715
	**1 or more**	17	77.3%	21	84%	0.715
	**2 or more**	10	45.5%	15	60%	0.319

Exacerbation defined as increase in cough, wheeze or shortness of breath. Upper respiratory tract infection defined as a cold (runny/congested nose, sore throat, earache, sneezing, with or without fever). Exacerbations (1): Number of subjects who had asthma exacerbation alone, without any sign of respiratory infection. Upper respiratory tract infection (2): Number of subjects who had URTI alone with no change in asthma symptoms. Exacerbations with URTI (3): Number of subjects who had asthma exacerbation with suspected URTI symptoms. Difference between groups analysed using Pearson’s Chi-squared test/Fisher’s exact test (expected cell values < 5) where appropriate. Values in bold indicate statistically significant results.

**Table 4 nutrients-14-03087-t004:** Severity and duration of Asthma exacerbation (with and without upper respiratory tract infections) reported by parent during the 6-month intervention using the Asthma flare-up diary for young children (ADYC) ^a^.

	Intervention (n = 18)		Control (n = 18)		
Efficacy Outcomes	Events (Number)	Median (IQRI)	Events (Number)	Median (IQR)	*p*-Value
**Asthma symptoms severity/event**					
ADYC cumulative daily scores ^b^	34	9.9 (5.2, 11.8)	37	9.8 (6.9, 12.8)	0.687
**Exacerbation duration**					
ADYC duration/event (days) ^c^	34	4.0 (3.0, 7.0)	37	4.0 (3.0, 5.0)	0.522
**β2-agonist use**					
ADYC cumulative number of puffs/event ^d^	33	42.0 (30.0, 92.0)	32	48.5 (27.0, 72.5)	0.526
**OCS courses/child**	8	1.0 (1.0, 2.0)	14	1.5 (1.0, 2.0)	0.462

^a^ Values for each group are reported as median (interquartile range (IQR)), *p*-values for difference between groups in symptom severity, duration and B^2^-agonist use calculated using linear mixed model adjusting for maintenance ICS use, accounting for repeated measures; *p*-values for difference between groups in number of OCS courses calculated using Wilcoxon rank sum tests. The number of events for which valid and complete ADYCs were completed is indicated each outcome. Subjects who did not return valid ADYCs were excluded from this analysis. ^b^ Measured on the 17-item ADYC [28], on a scale of 1 (best) to 7 (worst), completed daily for the duration of exacerbations. The asthma symptoms severity is the sum of the daily ADYC scores per exacerbation event. The ADYC items included cough (n = 2), wheezing (n = 2), dyspnea (n = 4), night awakenings (n = 1), general wellbeing (n = 5), and child’s response to salbutamol inhalations (n = 3). ^c^ Duration from the first day with two or more asthma symptoms to the last day with one or more asthma symptoms (cough, wheezing, and/or dyspnea) as indicated by ADYC. Up to one day without asthma symptoms could be included in an exacerbation. ^d^ Cumulative number of inhalations during asthma exacerbations, as indicated on the ADYCs.

**Table 5 nutrients-14-03087-t005:** qPCR for common respiratory viruses in asthma exacerbations and upper respiratory tract infections in children with asthma during 6-month dietary intervention study.

Event Type		Intervention	Control	*p*-Value
**Total Events Tested for Virus (1 + 2 + 3)**		45	56	
PCR virus-positive events (n)		16 (35.6%)	21 (37.5%)	0.840
Number of events positive for each respiratory virus	Rhinovirus	10 (22.2%) *	16 (28.6%) *	0.528
	Coronavirus	1 (2.2%)	0	
	RSV ^a^ A	0	1 (1.8%) *	
	RSV B	1 (2.2%)	3 (5.3%) *	
	Influenza A	2 (4.4%) *	5 (9.0%) *	
	Influenza B	3 (6.6%)	1 (1.8%) *	
**Exacerbations (1) tested for virus**		5	6	
PCR virus-positive events (n)		2 (40.0%)	2 (33.3%)	1.000
Number of events positive for each respiratory virus	Rhinovirus	1 (20.0%)	1 (16.7%)	1.000
	Coronavirus	0	0	
	RSV ^a^ A	0	1 (16.7%)	
	RSV B	0	0	
	Influenza A	0	0	
	Influenza B	1 (20.0%)	0	
**URTI (2) tested for virus**		11	11	
PCR virus-positive events (n)		4 (36.4%)	4 (36.4%)	1.000
Number of events positive for each respiratory virus	Rhinovirus	2 (18.2%) *	4 (36.4%) *	0.610
	Coronavirus	1 (9.1%)	0	
	RSV A	0	0	
	RSV B	0	0	
	Influenza A	2 (18.2%) *	1 (9.1%) *	
	Influenza B	0	1 (9.1%) *	
**Exacerbation with URTI (3) tested for virus**		29	39	
PCR virus-positive events (n)		10 (34.5%)	15 (38.5%)	0.597
Number of events positive for each respiratory virus	Rhinovirus	7 (24.1%)	11 (28.2%) *	0.120
	Coronavirus	0	0	
	RSV A	0	0	
	RSV B	1 (3.4%)	3 (7.7%) *	
	Influenza A	0	4 (10.2%) *	
	Influenza B	2 (6.8%)	0	

Data are presented as n (%). Exacerbations defined as increase in cough, wheeze or shortness of breath. Upper respiratory tract infection defined as a cold (runny/congested nose, sore throat, earache, sneezing, with or without fever). Exacerbations (1): Number of times that parent reported asthma exacerbation alone, without any sign of respiratory infection. Upper respiratory tract infection (2): Number of times that parent reported URTI alone with no change in asthma symptoms. Exacerbations with URTI (3): Number of times that parent reported asthma exacerbation with suspected URTI symptoms. ^a^ Respiratory syncytial virus; * Multiple-virus positive event(s). Difference between groups analysed using Pearson’s Chi-squared test/Fisher’s exact test (expected cell values < 5) where appropriate.

**Table 6 nutrients-14-03087-t006:** Cytokine secretion of PBMCs from children with asthma in response to Rhinovirus-1B, House Dust Mite and Lipopolysaccharide.

Variable	Baseline ^a^	3 Months ^a^	Adjusted Change ^b^ Coeff. [95% CI]	*p*-Value	6 Months ^a^	Adjusted Change ^b^ Coeff. [95% CI]	*p*-Value
**Levels of cytokines in the supernatants of PBMCs stimulated with Rhinovirus-1B**							
IFN-γ (pg/mL)							
**Intervention**	19.62 (9.37, 32.39)	20.21 (7.92, 46.16)	4.72 [−11.08, 20.53]	0.558	16.64 (7.87, 43.66)	3.84 [−16.65, 24.34]	0.713
**Control**	14.08 (7.38, 46.20)	18.38 (11.38, 37.51)			15.29 (7.20, 32.26)		
IFN-λ (pg/mL)							
**Intervention**	3.77 (1.66, 4.74)	2.57 (1.76, 5.08)	−1.58 [−28.06, 24.35]	0.890	2.14 (1.18, 6.42)	15.34 [−10.68, 41.38]	0.248
**Control**	6.68 (4.08, 39.29)	5.72 (1.43, 33.27)			5.5 (1.77, 15.85)		
IL-1β (pg/mL)							
**Intervention**	22.47 (3.78, 71.17)	41.92 (5.83, 77.87)	−60.53 [−167.30, 46.24]	0.267	17.73 (4.41, 59.99)	−25.40 [−130.92, 80.11]	0.637
**Control**	21.25 (7.44, 92.67)	20.64 (7.93, 83.64)			9.36 (4.05, 42.13)		
IL-6 (ng/mL)							
**Intervention**	1.55 (0.13, 8.91)	3.24 (0.68, 12.62)	−2.01 [−14.91, 10.88]	0.760	1.45 (0.29, 6.38)	−1.56 [−14.32, 11.19]	0.810
**Control**	1.26 (0.38, 9.38)	1.19 (0.48, 10.32)			0.59 (0.21, 1.74)		
**Levels of cytokines in the supernatants of PBMCs stimulated with House Dust Mite**							
IFN-γ (pg/mL)							
**Intervention**	1.02 ± 0.37	1.25 ± 0.39			1.03 ± 0.48		
**Control**	0.98 ± 0.56	0.98 ± 0.48	0.23 [−0.14, 0.60]	0.221	0.90 ± 0.49	0.00 [−0.36, 0.36]	0.999
IFN-λ (pg/mL)							
**Intervention**	2.18 (0.82, 3.29)	2.91 (1.57, 5.08)	−1.89 [−11.50, 7.71]	0.699	1.67 (0.94, 5.75)	0.49 [−11.81, 12.80]	0.937
**Control**	2.17 (1.26, 5.94)	2.88 (1.11, 7.89)			2.29 (0.92, 6.40)		
IL-1β (pg/mL)							
**Intervention**	13.89 (7.29, 42.49)	23.65 (7.58, 57.85)	13.81 [−88.27, −115.91]	0.791	31.36 [5.80, 65.04)	5.54 [−95.45, 106.54]	0.914
**Control**	14.71 (10.49, 54.65)	22.38 (9.04, 42.47)			24.91 [9.26, 78.49)		
IL-6 (ng/mL)							
**Intervention**	1.87 (0.74, 7.42)	4.66 (0.75, 15.19)	1.15 [−13.90, 16.20]	0.881	2.20 (0.85, 15.12)	0.96 [−13.91, 15.83]	0.899
**Control**	5.56 (1.46, 12.78)	4.52 (1.04, 13.82)			10.11 (1.49, 20.10)		
**Levels of cytokines in the supernatants of PBMCs stimulated with Lipopolysaccharide**							
IFN-γ (pg/mL)							
**Intervention**	1.48 (1.07, 4.92)	**3.59 (1.37, 12.65) ^c^**			2.66 (1.35, 8.41)		
**Control**	2.37 (1.08, 13.65)	3.53 (1.56, 8.82)	−4.21 [−16.23, 7.81]	0.492	5.52 (0.73, 12.74)	−5.71 [−17.75, 6.32]	0.352
IFN-λ (pg/mL)							
**Intervention**	1.50 (0.6, 4.11)	**1.97 (0.64, 5.9) ^c^**	0.61 [−6.89, 8.12]	0.872	1.86 (0.4, 4.53)	4.47 [−4.71, 13.66]	0.340
**Control**	4.22 (0.93, 10.12)	0.92 (0.2, 1.99)			2.06 (0.63, 6.79)		
IL-1β (pg/mL)							
**Intervention**	1926.13 (1213.19, 4066.73)	4646.56 (1773.93, 6439.98)	997.36 [−1521.98, 3516.70]	0.438	1792.86 (1236.9, 4634.93)	−201.07 [−2663.57, 2261.42]	0.873
**Control**	1928.85 (1099.88, 8469.68)	1896.76 (1623.12, 9045.34)			2000.26 (968.92, 8832.58)		
IL-6 (ng/mL)							
**Intervention**	70.04 (46.64, 103.49)	76.63 (38.87, 150.94)	20.14 [−18.31, 58.59]	0.305	78.34 (52.04, 114.49)	−2.33 [−40.56, 35.88]	0.905
**Control**	75.83 (48.56, 130.21)	65.34 (46.23, 94.99)			88.99 (71.16, 151.03)		

Intervention (n = 22), Control (n = 24). All variables adjusted for media. ^a^ Unadjusted mean ± SD or median (IQR) are presented for baseline, 3 month and 6 month measures. ^b^ Adjusted change means and 95% confidence intervals [CI] are the differences in change from baseline in intervention group compared to control group by linear mixed model adjusted for random effects and hospital admissions 12 months prior to recruitment. ^c^
*p* < 0.05 difference within group compared to baseline or 3 month. Values in bold indicate statistically significant results.

## Data Availability

The datasets used and analysed during the current study are available from the corresponding author on reasonable request.

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
