# Peer review of "The Effects of Increasing Fruit and Vegetable Intake in Children with Asthma on the Modulation of Innate Immune Responses"

_nutrients, 2022, doi:10.3390/nu14153087_

Round 1

Reviewer 1 Report

Thank you for the opportunity to review the manuscript titled, The effects of increasing fruit and vegetable intake in children with asthma on the modulation of innate immune responses. In this randomized trial of children with asthma aged 3-11 years, those who consumed high fruits and vegetables for 6 months had reduced asthma-related illnesses compared to children who maintained a usual diet over the same period. This was a highly interesting manuscript, although some concerns remain.

1.       Please provide the reader with more detail regarding the location of the two hospitals at which recruitment took place.

2.       Data were collected between September 2015 and July 2018, and are thus 4 years old. It is unclear why there was a delay in reporting these findings. Although COVID-related interruptions may partly explain this delay, the data were collected 1.75 years before the pandemic started.

3.       How was “stable asthma” operationalised (Line 82)?

4.       It was pleasantly suprising to learn that participants were provided with food hampers (Lines 91-2). Please provide more detail here.

5.       Attrition was quite high, at 23.9%, or 16/67 (Line 170-1). Can the authors broadly explain reasons for this?

6.       There is no mention about ethnicity or socio-economic status? But, such descriptive results would help the reader to critically interpret the findings.

7.       Please describe access to medications. Was such access comparable between groups? Such information is important to understand the results.

8.       As mentioned in Comment 2, these data are now 4 years old. Since the data were collected, food prices have increased dramatically, which has resulted in forced choice between food purchases or medication purchases for some families. In this light, please discuss your intervention. Consider topics such as generalizability, sustainability and the need for programs and policy.

9.       Please use person-centric language, rather than disease-centric language (e.g. children with asthma, rather than (as noted in Line 18), asthmatic children.

Author Response

Response to Reviewer 1

Thank you very much for your close attention to our submitted manuscript. The comments have been very useful in improving the manuscript. Following are our responses.

  1. Please provide the reader with more detail regarding the location of the two hospitals at which recruitment took place.

Action or explanation: Thank you so much for your comment. Additional information regarding the location of the two recruitment sites have been added as requested (Lines 78- 9).

Lines 77-80: Participants were recruited via attendance to the emergency department or admission to the John Hunter Children’s Hospital, Newcastle, Australia and Maitland Hospital, Newcastle, Australia, following an asthma exacerbation, from September 2015 to July 2018.

  1. Data were collected between September 2015 and July 2018, and are thus 4 years old. It is unclear why there was a delay in reporting these findings. Although COVID-related interruptions may partly explain this delay, the data were collected 1.75 years before the pandemic started.

Action or explanation: Thank you so much for your comment. We agree with this comment and acknowledge that there was a delay in reporting these findings. This is partly because laboratory analysis performed as part of PhD thesis, which was successfully defended in June 2020. Additionally, the present study reports secondary findings from our RCT and our primary outcomes paper was published in July 2021. While our data were collected 4 years ago, mechanisms of viral induced asthma and effects of dietary intake on immune responses to viral infection have not changed, thus the paper is still highly relevant to readers.

  1. How was “stable asthma” operationalised (Line 82)?

Action or explanation: Thank you so much for your valuable suggestion. Stable asthma has been defined as requested (Lines 83-5).

Lines 83-5: stable asthma at the initial clinic visit (defined as no change in asthma medications, unscheduled medical visit for asthma, use of OCS or antibiotics in preceding 4 weeks).

  1. It was pleasantly surprising to learn that participants were provided with food hampers (Lines 91-2). Please provide more detail here.

Action or explanation: We appreciate this comment. Additional details regarding food hampers are added as requested (Lines 94- 9).

Lines 94- 9: the high F&V group received fortnightly food hampers, which included a selection of fresh and frozen fruit and vegetables, while the control group received monthly food baskets, which included a selection of carbohydrate-based foods, low in soluble fibre and antioxidants (bread, rice, pasta, cereal). The hampers were tailored to the subject's preferences, within the requirements of the dietary intervention.

  1. Attrition was quite high, at 23.9%, or 16/67 (Line 170-1). Can the authors broadly explain reasons for this?

Action or explanation: We acknowledge that our trial was limited by the high attrition rate (Intervention: 9/33 or 27.2%; Control: 9/34 or 26.4). This could be related to the parent burden, including time and commitment associated with study visit attendance, completing study questionnaires and participation in phone consultations and/ or child discomfort during blood collection. As suggested, we have explained this in the Discussion section of the manuscript (Lines 397- 400).

Lines 397- 400: Furthermore, the attrition rate (26.8%) was high, which may be related to the parent burden, including time and commitment associated with study visit attendance, completing study questionnaires and participation in phone consultations and/ or child discomfort during blood collection.

  1. There is no mention about ethnicity or socio-economic status? But, such descriptive results would help the reader to critically interpret the findings.

Action or explanation: Thank you so much for your precious suggestion. We have added ethnicity to the Table 1.

As shown below, the majority of participants in our study were Caucasian (Intervention: 77.3%; Control: 80.0%):

Ethnicity

Intervention (n=22)

Control (n=25)

P value

Caucasian

17 (77.3%)

20 (80.0%)

0.303

Aboriginal

0

2 (8.0%)

Latin American

0

1 (4.0%)

Asian

1 (4.5%)

0

Mixed

4 (18.2%)

2 (8.0%)

  1. Please describe access to medications. Was such access comparable between groups? Such information is important to understand the results.

Action or explanation: Thank you so much for your comment. Whilst we did not specifically collect information on medication access, the study was conducted in Australia, and it has been reported that Australia has achieved universal coverage of essential medicines (Morgan et al, Cost-related non-adherence to prescribed medicines, BMJ Open 2017). Therefore, we can confirm that access to prescriptions were comparable between the two groups.

  1. As mentioned in Comment 2, these data are now 4 years old. Since the data were collected, food prices have increased dramatically, which has resulted in forced choice between food purchases or medication purchases for some families. In this light, please discuss your intervention. Consider topics such as generalizability, sustainability and the need for programs and policy.

Action or explanation: Thank you so much for your suggestion. We have described the potential factors associated with a low F&V consumption, and highlighted the need for applying programs and policies that would help to increase F&V consumption (Lines 410-414).

Lines 410-14: While cost may be a significant barrier to fruit and vegetable consumption, other factors such as taste, preference, and culture can also significantly impact F&V purchase and consumption. There is growing evidence to indicate that participation in programs such as community kitchens or school gardens can help to increase F&V consumption.

Action or explanation: Thank you so much for your comment. As indicated above, our study was conducted in Australia, where prescription drug coverage is readily available with minimal direct cost to patients. Therefore, rising food prices would not necessarily affect adherence to medications. However, we agree that strategies to ensure that fruit and vegetables are widely available and affordable are important.

  1. Please use person-centric language, rather than disease-centric language (e.g., children with asthma, rather than (as noted in Line 18), asthmatic children.

Action or explanation: We agree and have changed the manuscript accordingly.

Reviewer 2 Report

In this study, the authors investigated the effect of increasing the intake of fruit and vegetable on the innate immune responses in children with asthma. The authors have shown that a diet high in fruit and vegetable (F&V) reduces asthma exacerbations in children with enhanced cytokine response to ex-vivo stimulation. The findings are very interesting and support the recent evidence that a healthy diet rich in F&V reduces asthma symptoms and supports greater asthma control. The data is nicely presented.

Comment:

One of the key results presented in this paper is that “the high F&V intervention increased LPS-induced interferon (IFN)-γ and IFN-λ production after 3 months compared to baseline (P<0.05)”. The concentrations of these cytokines were very low, close to the limit of detection and the increase is marginal. Hence, a cautious approach should be applied to report this data and mention this limitation in the abstract and other relevant sections. Please consider concentrating the culture supernatants and repeating the cytokine assay for better confidence in the result.

All the cytokines were measured 48 hours later. The higher dose of LPS used in this experiment will cause rapid cytokine secretion. Hence, analysis of cytokine response at 6 and 24 hours will be informative due to the temporal kinetics of these cytokines.

A good PBMC preparation has a lower percentage of monocytes and other myeloid cells capable of producing IFN-γ and IFN-λ. Is the lower level of these cytokines due to a lower percentage of myeloid cells in the PBMC?

Author Response

Response to Reviewer 2

Thank you very much for your close attention to our submitted manuscript. The comments have been very useful in improving the manuscript. Following are our responses.

  1. One of the key results presented in this paper is that “the high F&V intervention increased LPS-induced interferon (IFN)-γ and IFN-λ production after 3 months compared to baseline (P<0.05)”. The concentrations of these cytokines were very low, close to the limit of detection and the increase is marginal. Hence, a cautious approach should be applied to report this data and mention this limitation in the abstract and other relevant sections. Please consider concentrating the culture supernatants and repeating the cytokine assay for better confidence in the result.

Action or explanation: Thank you so much for your comments. We acknowledge that the rise in LPS-induced IFN-γ and IFN-λ production is marginal and have addressed this in the abstract and the results/ discussion, as requested.

Lines 25-6: LPS-induced interferon (IFN)-γ and IFN-λ production showed a small but significant  increase in the high F&V group after 3 months compared to baseline (P<0.05).

Lines 278- 9: However, after 3 months, LPS-induced IFN-γ and IFN-λ production showed a small but significant increase in the intervention group compared with baseline.

Moreover, in our previous case-control study, we observed a deficient IFN-γ response to LPS in asthmatic children compared with age-matched healthy controls (Hosseini et al, Children with Asthma Have Impaired Innate Immunity and Increased Numbers of Type 2 Innate Lymphoid Cells Compared With Healthy Controls. Front Immunol. 2021 Jun 17;12:664668). Similarly, Contoli et al. showed primary bronchial epithelial cells and alveolar macrophages from adult asthmatic patients produced lower levels of IFN-λ following LPS stimulation than healthy controls (Contoli M, et al. Role of deficient type III interferon-lambda production in asthma exacerbations. Nature medicine 2006;12(9):1023-6). 

We have also addressed this in the discussion section (Lines 353- 7).

Unfortunately, we are not able to perform additional analysis due to insufficient specimen volume. Whilst concentrating the samples would likely increase the concentrations of these cytokines, similar effects would be expected. .

  1. All the cytokines were measured 48 hours later. The higher dose of LPS used in this experiment will cause rapid cytokine secretion. Hence, analysis of cytokine response at 6 and 24 hours will be informative due to the temporal kinetics of these cytokines.

Action or explanation: The concentrations LPS (100ng/ml) was selected using previously published literature (Reference: Plevin et al. The Role of Lipopolysaccharide Structure in Monocyte Activation and Cytokine Secretion.” Shock (Augusta, Ga.)2016).

In terms of timing, since we aimed to look at protein expression in PBMCs supernatants, 6hrs would not be sufficient. We agree that 24hours would provide interesting information, however we did not have enough PBMCs to do both timepoints.  We have addressed this limitation in the discussion section (Lines 402- 5).

Lines 402- 5: Finally, due to insufficient PBMC samples, we could not determine whether cytokines responses would differ at 24 hours or 72 hours. However, a deficient interferon response to RV in asthmatic children was also reported in a previous study that harvested supernatants at 72 hours (Iikura K, et al. Peripheral blood mononuclear cells from patients with bronchial asthma show impaired innate immune responses to rhinovirus in vitro. Int Arch Allergy Immunol 2011;155 Suppl 1:27-33).

  1. A good PBMC preparation has a lower percentage of monocytes and other myeloid cells capable of producing IFN-γ and IFN-λ. Is the lower level of these cytokines due to a lower percentage of myeloid cells in the PBMC?

Action or explanation:

Action or explanation:  Thank you for your comment. The methodology for PBMC isolation and culture was obtained from “Forbes et al. Impaired type I and III interferon response to rhinovirus infection during pregnancy and asthma, Thorax, 2012”. Monocytes typically account for 10%-15% of total PBMCs found in our blood samples. This was comparable to what we found in blood samples of aged-matched healthy controls.

As previously described, our previous case- control study demonstrated that children with asthma had deficient IFN-γ production in response to both RV1B and LPS infection, compared with the healthy controls. Moreover, both IL-1β and IL-6 production were significantly lower in response to HDM and LPS in children with asthma (Hosseini et al, Children with Asthma Have Impaired Innate Immunity and Increased Numbers of Type 2 Innate Lymphoid Cells Compared with Healthy Controls. Front Immunol. 2021 Jun 17;12:664668). Thus, we hypothesise that the impaired inflammatory response observed in patients with asthma is due to an imbalance between type I and type II cytokines. We have also addressed this in the discussion section (Lines 353- 7).

Lines 353-7: We have previously shown that PBMCs from children with asthma had deficient IFN-γ production in response to both RV1B and LPS infection compared with healthy controls(39). Other studies have also revealed that children with asthma, compared with age-matched controls, had defective or impaired IFN responses to respiratory viruses(40, 41).

Round 2

Reviewer 1 Report

Thank you for your careful attention to my original comments. At this time, I have some additional comments:

1 Table 1: The term "Caucasian" refers to individuals from the Caucasus region between the Black Sea and Caspian Sea. Do the authors mean "White" ? The authors may also wish to consider if they mean ethnicity here ("the ethnic or cultural origins of the person's ancestors") or race.

2 Table 2: The authors use "gender" then describe by male to female ratio. When describing males and females, this description is, in fact, sex, not gender. Either the authors report on sex, or on gender (and the appropriate descriptors, of which there are more than 2). The authors could also report on both sex, and on gender. However, these need to be reported separately, as the terms are not interchangeable.  

3 Lines 397-400: It is interesting to read that the authors interpreted the attrition to parental burden, particularly in light of the provision of food hampers. Was attrition explored qualitatively, or did the authors simply speculate the reasons for attrition?

4 Lines 412-414: This is not a conclusion, but rather belongs much earlier in the discussion. This statement must also be supported with references.

Author Response

Response to Reviewer 1

Thank you very much for your close attention to our submitted manuscript. The comments have been very useful in improving the manuscript. Following are our responses.

1 Table 1: The term "Caucasian" refers to individuals from the Caucasus region between the Black Sea and Caspian Sea. Do the authors mean "White" ? The authors may also wish to consider if they mean ethnicity here ("the ethnic or cultural origins of the person's ancestors") or race.

Action or explanation: Thank you so much for bringing this to our attention. The word “Caucasian” was actually used as a description of white race in our study. Therefore, we have replaced the term “Caucasian” with “White” and “ethnicity” with “race”.

2 Table 1: The authors use "gender" then describe by male to female ratio. When describing males and females, this description is, in fact, sex, not gender. Either the authors report on sex, or on gender (and the appropriate descriptors, of which there are more than 2). The authors could also report on both sex, and on gender. However, these need to be reported separately, as the terms are not interchangeable.  

Action or explanation: We agree and have replaced the term “gender” with “Sex” in Table 1.

3 Lines 397-400: It is interesting to read that the authors interpreted the attrition to parental burden, particularly in light of the provision of food hampers. Was attrition explored qualitatively, or did the authors simply speculate the reasons for attrition?

Action or explanation: Thank you for your comment. Reasons for wishing to discontinue the study were given by parents who withdrew.

4 Lines 412-414: This is not a conclusion, but rather belongs much earlier in the discussion. This statement must also be supported with references.

Action or explanation: We agree and have moved this statement to the discussion.

Authors were asked not to modify the Reference section, so reference for each statement was noted in the comment section and will be added to the manuscript by the editorial team if the manuscript gets accepted.

Line 396- 402: While cost may be a significant barrier to fruit and vegetable consumption, other factors such as taste, preference, and culture can also significantly impact F&V purchase and consumption. There is growing evidence to indicate that participation in incentive programs such as community kitchens or school gardens can help promote consumption of F&V. Parental and children’s knowledge of the protective effect of the F&V consumption on asthma is of great importance and need to be included in the management of children with asthma. 

Reviewer 2 Report

The authors have addressed the queries satisfactorily.